# Lasting antibody and T cell responses to SARS-CoV-2 in COVID-19 patients three months after infection

Xiao-Lin Jiang [1,7], Guo-Lin Wang[2,7], Xiang-Na Zhao[3,7], Fei-Hu Yan[4,7], Lin Yao[2,7], Zeng-Qiang Kou[1], Sheng-Xiang Ji[5], Xiao-Li Zhang[5], Cun-Bao Li[5], Li-Jun Duan[2], Yan Li[1], Yu-Wen Zhang[1], Qing Duan[1], Tie-Cheng Wang[4], En-Tao Li[4], Xiao Wei[3], Qing-Yang Wang[6], Xue-Feng Wang[4], Wei-Yang Sun[4], Yu-Wei Gao[4,8], Dian-Min Kang [1,8], Ji-Yan Zhang [6,8✉] & Mai-Juan Ma [2,8✉]

The dynamics, duration, and nature of immunity produced during SARS-CoV-2 infection are still unclear. Here, we longitudinally measured virus-neutralising antibody, specific antibodies against the spike (S) protein, receptor-binding domain (RBD), and the nucleoprotein (N) of SARS-CoV-2, as well as T cell responses, in 25 SARS-CoV-2-infected patients up to 121 days post-symptom onset (PSO). All patients seroconvert for IgG against N, S, or RBD, as well as IgM against RBD, and produce neutralising antibodies (NAb) by 14 days PSO, with the peak levels attained by 15–30 days PSO. Anti-SARS-CoV-2 IgG and NAb remain detectable and relatively stable 3–4 months PSO, whereas IgM antibody rapidly decay. Approximately 65% of patients have detectable SARS-CoV-2-specific CD4$^+$ or CD8$^+$ T cell responses 3–4 months PSO. Our results thus provide critical evidence that IgG, NAb, and T cell responses persist in the majority of patients for at least 3–4 months after infection.

[1] Shandong Provincial Center for Disease Control and Prevention, Jinan, China. [2] State Key Laboratory of Pathogen and Biosecurity, Beijing Institute of Microbiology and Epidemiology, Beijing, China. [3] Institute of Disease Control and Prevention, Chinese People's Liberation Army, Beijing, China. [4] Institute of Military Veterinary Medicine, Academy of Military Medical Sciences, Changchun, China. [5] Linyi Center for Disease Control and Prevention, Linyi, China. [6] Beijing Institute of Brain Science, Beijing, China. [7] These authors contributed equally: Xiao-Lin Jiang, Guo-Lin Wang, Xiang-Na Zhao, Fei-Hu Yan, Lin Yao. [8] These authors jointly supervised: Yu-Wei Gao, Dian-Min Kang, Ji-Yan Zhang, Mai-Juan Ma. ✉email: jiyanzhang@hotmail.com; mjma@163.com

   1

Since its emergence in December 2019, the severe acute respiratory syndrome coronavirus 2 (SARS-CoV-2) that causes coronavirus disease 2019 (COVID-19) has caused a pandemic affecting 216 countries, areas, or territories with > 13 million cases and at least 574,464 deaths as of July 15, 2020[1]. In China, the number of laboratory-confirmed cases has exceeded 80,000, resulting in >4000 deaths[1]. Populations immuno-naive to SARS-CoV-2 are considered to have markedly contributed to the dramatic increase in cases worldwide. Transmission modelling studies of SARS-CoV-2 assume that infection produces immunity to reinfection for durations of at least one year[2,3]. The dynamics and maintenance of immunity and the nature of the protection it affords are critical for serologic diagnosis, the therapeutic use of convalescent sera, population-based sero-epidemiological surveys, vaccine design and development, and vaccination strategies.

Several studies have reported that patients infected with SARS-CoV-2 infection can produce an antibody response[4–8], but reported information had focused primarily on hospitalised patients where only virus-specific IgG and IgM antibodies were measured. Moreover, despite the thorough characterisation of many monoclonal neutralising antibodies isolated from COVID-19 patients or animal models[9–16], the polyclonal serum neutralising antibody induced after natural infection—which is important for virus clearance and has been considered a critical immune product for protection against virus infection—has been evaluated in limited studies[5,7,8,17]. Wang et al. found all patients to be seropositive for IgG and neutralising antibodies 41–53 days after illness onset[5]. Furthermore, Ni and colleagues detected neutralising antibodies (in all but one patient) and high titre for IgG antibody in all patients, either newly discharged or two weeks after discharge[7]. Surprisingly, Wu et al. reported the detection of low neutralising antibodies or none, in some hospitalised patients 2–3 weeks post-symptom onset (PSO)[6]. However, in a recent study, Long et al. found that while neutralising antibodies were detectable in all patients eight weeks after discharge, the measured titre had decreased significantly, and nearly 13% of the symptomatic patients became negative for IgG in the early convalescent phase[8]. These data raise questions about protective immunity and the appropriate amount of time that should be recommended for quarantine[18,19].

Some studies have already pointed to T cells as the potential key to solving this dilemma. SARS-CoV-2-specific memory T cell phenotypes (central memory for CD4 and effector memory for CD8 lymphocytes) were observed in the peripheral blood of one case-patient two weeks PSO[20]. Additionally, several studies reported the detection of SARS-CoV-2-specific CD8+ and CD4+ T cells in the majority of COVID-19 convalescent patients approximately 3–5 weeks PSO[7,21–23]. This process may be capable of providing useful information about protective immunity. However, to date, information regarding the dynamics, duration, and nature of immunity produced during SARS-CoV-2 infection is limited.

Here, we longitudinally assess SARS-CoV-2-infected patients up to 3–4 months post-infection and analysed their SARS-CoV-2-specific antibody and memory T cell responses over time. We find that SARS-CoV-2-specific antibody and T cell responses maintain in most recovered patients for at least 3–4 months after infection.

## Results

**Characteristic of patients and samples**. We enroled 25 laboratory-confirmed SARS-CoV-2 patients, of which 3 were severe patients, 18 were moderate patients, and 4 were asymptomatic (Table 1). Their median age was 40 years (interquartile range [IQR], 33–53), and 13 (52%) were male. The most commonly reported symptoms were fever and cough. Seventy-two percent of patients experienced moderate illness. Fifty-two percent of these individuals had known underlying medical illnesses. In the time between PSO and sampling, we collected a total of 112 serum samples. We collected serum at multiple time points for most patients (n = 16, 64%), collecting having ≥ 6 serum samples with 56% (n = 14) (Table 1). We collected 21 serum samples from 12 patients ≤7 PSO, 30 from 16 patients ≤14 days PSO, 19 from 11 patients ≤21 days PSO, 22 from 16 patients 28 days PSO, and 20 from 20 patients 3–4 months PSO.

**Detection of virus-specific IgG and IgM in patients**. The two structural proteins of SARS-CoV-2, nucleocapsid (N) and spike (S) protein, have been used as target antigens for serological assays. Although it is unlikely that antibody responses to N protein can directly neutralise SARS-CoV-2, this is the antigen targeted by multiple commercial assays. Therefore, to study antibody responses to SARS-CoV-2, we first qualitatively measured IgG against N and IgM against the receptor-binding domain (RBD) of the SARS-CoV-2 S protein in serum from patients at a 1:10 dilution using two well-validated commercial diagnostic ELISA kits[4]. Four (33.3%) of 12 patients with serum samples collected from 1 to 7 days PSO tested positive for anti-N IgG, and 4 (33.3%) patients tested positive for anti-RBD IgM (Fig. 1a–d). Of these, one patient was positive for both anti-N IgG and anti-RBD IgM antibodies. Of 16 patients with serum samples collected from 8-14 days PSO, 13 (81.3%) and 14 (87.5%) were positive for anti-N IgG and anti-RBD IgM antibodies, respectively. Thereafter, 100% of patients with serum samples collected >14 days PSO were positive for both anti-N IgG and anti-RBD IgM antibodies before discharge. All patients remained positive for anti-N IgG 3–4 months PSO (Fig. 1a, c), whereas 15 (75%) of 20 were positive for anti-RBD IgM antibodies, and the antibody levels rapidly decayed (Fig. 1c, d). By contrast, there were no detectable anti-N IgG or anti-RBD IgM antibodies among healthy controls (Fig. 1a, c).

Given that the neutralising antibody (NAb) response for SARS-CoV-2 primarily targets the S protein, using SARS-CoV-2-derived recombinant trimeric S protein and monomeric RBD, we then carried out ELISAs to quantitatively detect IgG antibody binding in serum. Similar to our observation of anti-N IgG and anti-RBD IgM responses, we detected IgG antibodies binding S (Fig. 1e–f; geometric mean endpoint titer 5,358; 95% confidence interval [CI], 3062.0–9375.6) or RBD (Fig. 1g–h; mean, 2056; 95% CI, 654.6–6471.4) in all patients >14 days PSO. Although there was a slight decline in antibody titre 3–4 months PSO, anti-S (mean 4345.1; 95%CI, 3097.4–6095.4) or anti-RBD (mean 1513.6; 95%CI, 635.3–3614.1) IgG antibodies remained detectable in all patients but one patient (patient 12). As expected, in the serum of healthy controls we observed a minimal reactivity of anti-S or anti-RBD IgG antibodies (Fig. 1e, g). Collectively, these findings indicate that COVID-19 patients produced IgG and IgM antibody responses to SARS-CoV-2, and that the IgG antibodies can persist at least 3–4 months PSO.

**Virus-specific neutralising antibody in patients**. We performed a live virus-based neutralising assay to further study NAb against SARS-CoV-2 in the serum, We found that 58.3% (7/12) of patients had positive NAb within one week PSO (Fig. 2a, b) with a geometric mean titer (GMT) of 41.3 (95%CI 24.9–68.7).Within 14 days PSO, the positivity rate increased to 93.8% (15/16), and thereafter all patients had positive NAb. We observed the highest GMT (1280, 95%CI 873.8–1875.0) in the sera of patients 15–21 days PSO, After which the GMT trend declined to 1165 (95%CI 824.1–1646.0) approximately one-month PSO, and 697.9

**Table 1 Clinical characteristics of COVID-19 patients.**

| Pt# | Age, y/sex | Symptoms[a] | Underlying disease | Days to admission[b] | Days in hospital | Days in ICU | CT findings of ground-glass opacities | Days of sample collection[c] | Disease severity |
|---|---|---|---|---|---|---|---|---|---|
| 1 | 25/M | 1, 2, 9 | FL | 2 | 27 | 5 | Both lungs | 4, 7, 9, 13, 16, 27, 106 | Severe |
| 2 | 64/M | 1, 2, 6, 7, 10 | RC | 4 | 19 | 4 | Both lungs | 6, 9, 12, 15, 19, 23, 37, 116 | Severe |
| 3 | 27/M | 1, 2, 7, 9 | DM, NEP | 0 | 23 | 7 | Both lungs | 1, 4, 7, 11, 15, 18, 29 | Severe |
| 4 | 51/M | 1, 2 | No | 4 | 17 | 0 | Both lungs | 1, 4, 6, 10, 13, 24, 103 | Moderate |
| 5 | 46/F | 1, 2 | AST, HTN | 0 | 15 | 0 | Both lungs | 1, 4, 6, 10, 13, 24, 103 | Moderate |
| 6 | 23/F | 1, 2 | No | 4 | 14 | 0 | Both lungs | 6, 9, 13, 16, 27, 106 | Moderate |
| 7 | 51/M | 1, 2 | HTN | 8 | 17 | 0 | Both lungs | 13, 15, 19, 22, 33, 112 | Moderate |
| 8 | 61/F | 1, 2, 4, 6, 7, 8, 10 | No | 9 | 16 | 0 | Both lungs | 11, 14, 18, 20, 22, 24, 42, 121 | Moderate |
| 9 | 36/M | 1, 2, 5, 6 | CB, FLUL | 0 | 13 | 0 | Both lungs | 2, 5, 8, 10, 12, 30, 109 | Moderate |
| 10 | 38/F | 1, 2 | HTA, PTC, HTM | 1 | 17 | 0 | Both lungs | 3, 8, 11, 15, 18, 29, 108 | Moderate |
| 11 | 41/M | 1, 2, 6 | No | 7 | 20 | 0 | Both lungs | 9, 15, 17, 21, 24, 35, 114 | Moderate |
| 12 | 34/M | 1, 2, 5 | No | 8 | 15 | 0 | Both lungs | 11, 14, 16, 20, 23, 34 | Moderate |
| 13 | 32/F | 1, 2, 6, 7, 8, 11 | No | 1 | 17 | 0 | Both lungs | 5, 8, 27 | Moderate |
| 14 | 46/F | 1, 2 | LT, HTM | 5 | 17 | 0 | Both lungs | 106 | Moderate |
| 15 | 54/M | 1, 2 | No | 0 | 15 | 0 | Both lungs | 97 | Moderate |
| 16 | 72/M | 1, 2, 5 | DM, HTN | 13 | 13 | 0 | Both lungs | 109 | Moderate |
| 17 | 40/M | 1, 2, 5 | No | 7 | 11 | 0 | Both lungs | 101 | Moderate |
| 18 | 40/F | 1 | CS | 11 | 14 | 0 | Both lungs | 98 | Moderate |
| 19 | 38/F | 1, 2 | No | 7 | 13 | 0 | Left lung | 103 | Moderate |
| 20 | 54/F | 1, 2 | LEU | 1 | 15 | 0 | Both lungs | 102 | Moderate |
| 21 | 61/F | 3 | CHD, HTN | 1 | 12 | 0 | Right lung | 92 | Moderate |
| 22 | 52/F | No | No | 0 | 21 | 0 | Both lungs | 2, 5, 8, 10, 12, 16, 30 | Asymptomatic |
| 23 | 33/F | No | No | 1 | 10 | 0 | Both lungs | 1, 3, 7, 8, 10, 28, 107 | Asymptomatic |
| 24 | 33/M | No | HH | 0 | 18 | 0 | Normal | 5, 8, 27 | Asymptomatic |
| 25 | 3/M | No | No | 1 | 14 | 0 | Normal | 98 | Asymptomatic |

[a]1, fever; 2, cough; 3, dry throat; 4, rhinorrhea; 5, chest tightness; 6, fatigue; 7, nausea; 8, vomiting; 9, diarrhoea; 10, abdominal discomfort; 11, bloody stools. [b]After symptom onset or initial positive RT-PCR results (for asymptomatic patients). AST asthma, CB chronic bronchitis, CHD coronary heart disease, CS caesarean section, DM diabetes mellitus, FL fatty liver, FLUL fracture of left upper limb, HTA hepatic hemangioma, HTA hysteromyoma, HTM hypothyroidism, HTN hypertension, LT lumbar tuberculosis, LEU leukopenia, NEP nephritis, PTC papillary thyroid carcinoma, RC renal calculus, HI hepatic insufficiency.

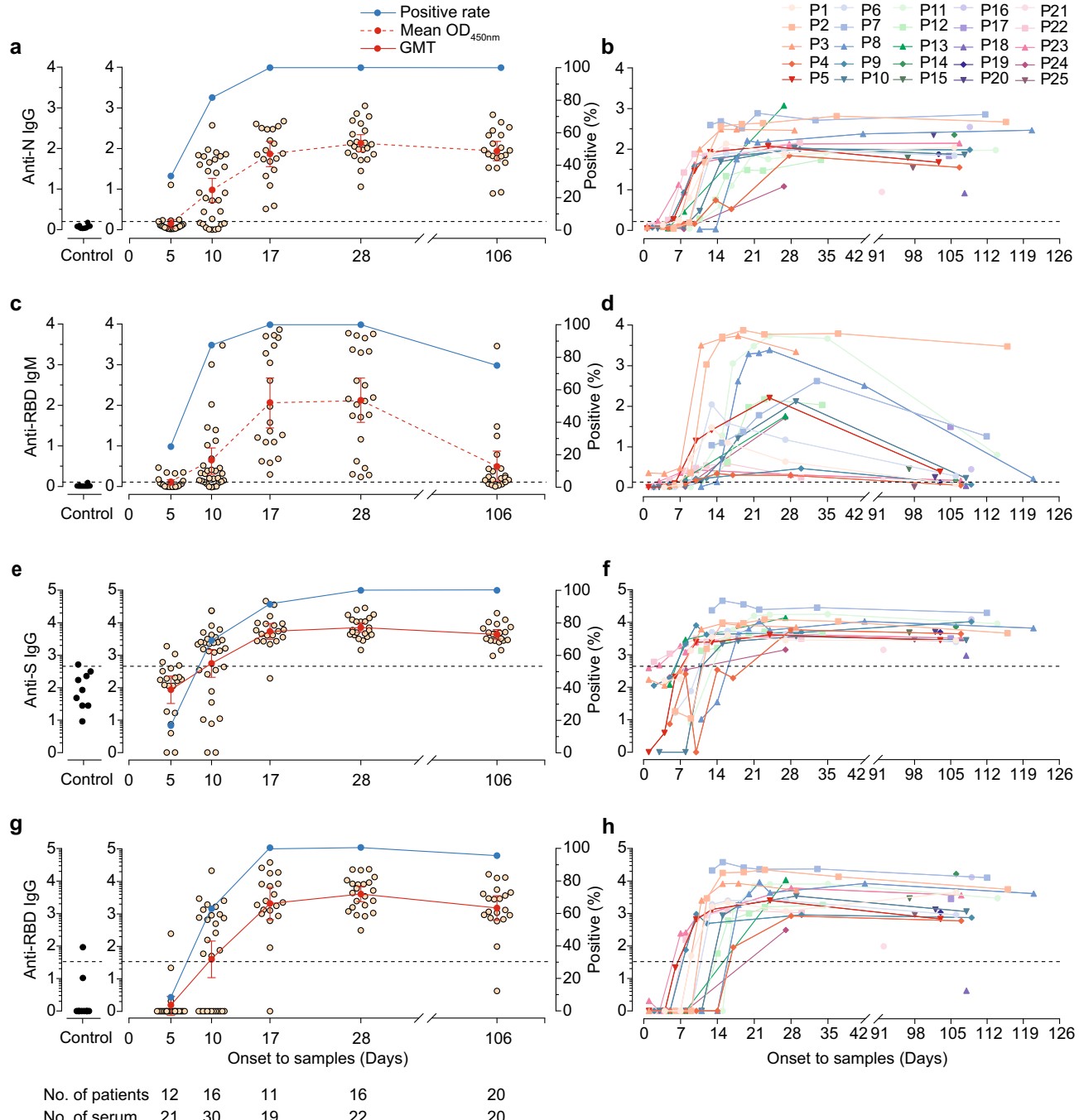

**Fig. 1 IgG and IgM antibody response kinetics in the serum of patients with SARS-CoV-2 infection, by days after symptom onset. a–d** The percentages (blue line) of patients (P) with serum samples that were positive for IgG to the nucleocapsid (N) protein (**a**), IgM to the receptor-binding domain (RBD) of SARS-CoV-2 spike (S) (**b**), IgG to the S (**c**) and RBD (**d**), and the corresponding mean optical density (OD) for anti-N IgG and anti-RBD IgM and $\log_{10}$-transformed geometric mean endpoint titer (GMT) for anti-S and -RBD IgG (red dashed line). Error bars represent the 95% confidence interval. Each circle represents the titer for a serum sample. **e–h** Individual level for anti-N IgG (**e**), anti-RBD IgM (**f**), anti-S IgG (**g**), and anti-RBD IgG (**g**) in serum samples collected from patients, and the samples from the same patients are connected by the lines. Black dashed line indicates the threshold for positivity (anti-N IgG = 0.19, anti-S IgG = 439.5, anti-RBD IgG = 33.2, and anti-RBD IgM = 0.105). Source data included as a Source Data File.

(95%CI 401.0–1215.0) 3–4 months PSO. Notably, 85% (17/20) of patients still had high NAb titre of ≥1:640 3–4 months PSO, whereas we observed relatively low NAb titre (≤1:80) for two moderate illness patients (patients 18 and 19) and one asymptomatic individual (patient 25, a 3-year-old boy). We detected no NAb in the serum from healthy controls. We observed a significant correlation between NAb titre and anti-N IgG (Spearman $r = 0.672$, $p < 0.0001$), anti-S IgG (Spearman $r = 0.668$,

$p < 0.0001$), and anti-RBD IgG (Spearman $r = 0.707$, $p < 0.0001$) (Fig. 2c). Anti-RBD IgM antibodies were also correlated with NAb titre (Spearman $r = 0.714$, $p < 0.0001$). We additionally observed a strong correlation between anti-S IgG and anti-RBD IgG antibodies (Fig. 2c; Spearman $r = 0.908$, $p < 0.0001$). These findings suggest that patients produced robust NAb responses after SARS-CoV-2 infection, and that the majority of patients' NAb titre persisted 3–4 months PSO.

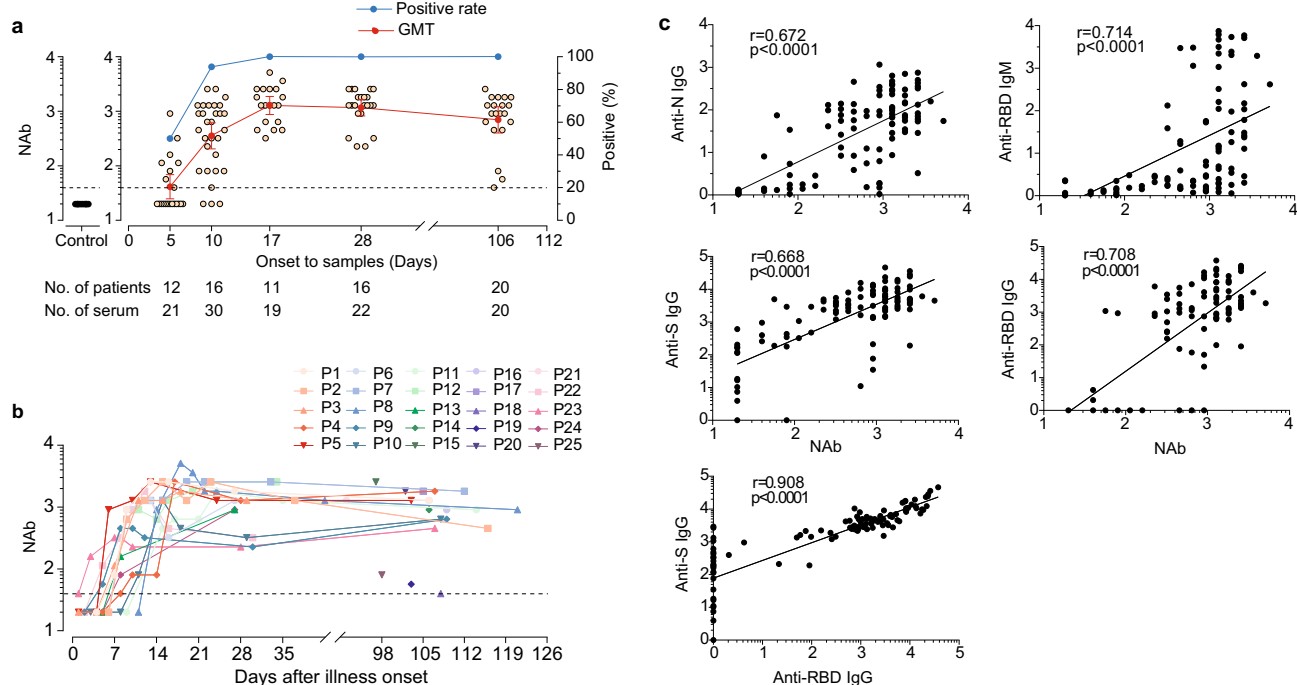

**Fig. 2 SARS-CoV-2 live-virus neutralisation antibody titre in serum and correlation with IgG and IgM responses. a** The percentages of patients with serum samples that were positive for neutralising antibody (NAb) and log10-transformed geometric mean titer (GMT). Error bars represent 95% confidence interval. Each circle represents the titer for a serum sample. **b** Individual NAb titer in serum samples collected from patients, and the samples from the same patients are connected by the lines. **c** Correlations between SARS-CoV-2-specific NAb titer and anti- nucleoprotein (N), -spike (S), and -receptor-binding domain (RBD) IgG levels and anti-RBD IgM, and the correlation between anti-S IgG and anti-S IgG. Statistical comparisons were performed using the two-sided nonparametric Spearman correlation. p value and spearman's rho are presented. p = 4.79E-16, 9.93E-19, 8.46E-16, and 2.76E-18 for anti-N IgG, anti-RBD IgM, anti-S IgG, and anti-RBD IgG with NAb correlation, respectively, and p = 1.78E-43 for anti-S IgG with anti-RBD IgG correlation. Source data included as a Source Data File.

**CD4$^+$ and CD8$^+$ T cell responses to SARS-CoV-2 in recovered patients**. To assess SARS-CoV-2-specific T cell responses, we used a recombinant replication-deficient adenovirus type 5 vector encoding the SARS-CoV-2 S (rAd5-S) protein or N (rAd5-N) protein with flow cytometry in a serial intracellular cytokine (IFN-γ, TNF-α, and GzmB) staining (ICS) assay with peripheral blood mononuclear cells (PBMCs) from 20 patients 3–4 months PSO (Fig. 3a). Because Ad5 can efficiently transduce many types of cells, including antigen presentation cells, we used rAd5-S and rAd5-N instead of peptides or proteins for these assays to better mimic the in vivo stimulation and avoid possible negligence of specific epitopes. As shown in Fig. 3b, we respectively detected CD4$^+$ T cells producing IFN-γ in response to rAd5-S and rAd5-N in 10 (50%) and 13 (~65%) of 20 recovered patients; in 10 of them we detected CD4$^+$ T cells producing IFN-γ in response to both rAd5-S and rAd5-N. For CD8$^+$ T cell responses, we detected CD8$^+$ T cells producing IFN-γ in response to rAd5-S in only 3 (15%) of 20 patients, while detecting rAd5-N in 10 (50%) patients. We detected CD4$^+$ and CD8$^+$ T cells producing IFN-γ in response to both rAd5-S and rAd5-N in only two patients. These results indicate that most of the recovered patients had detectable SARS-CoV-2-specific CD4$^+$ or CD8$^+$ T cell responses 3–4 months PSO. Additionally, rAd5-S typically elicited CD4$^+$ T cell responses, whereas rAd5-N elicited both CD4$^+$ and CD8$^+$ T cell responses.

We further measured TNF-α co-expression in all the 14 patients with IFN-γ$^+$ CD4$^+$ or CD8$^+$ T cells and co-expression of GzmB in 8 of them (Fig. 3c). In response to rAd5-S, we detected TNF-α co-expression with IFN-γ in 7 of the 10 patients with S-specific CD4$^+$ T cells and 2 of the 3 patients with S-specific CD8$^+$ T cells, while finding GzmB co-expression with

IFN-γ in 4 of the 6 patients examined with S-specific CD4$^+$ T cells and one patient examined with S-specific CD8$^+$ T cells (Fig. 3d, e). In response to rAd5-N, we detected TNF-α co-expression with IFN-γ in 11 of the 13 patients with N-specific CD4$^+$ T cells and 6 of the 7 patients with N-specific CD8$^+$ T cells, while finding GzmB co-expression with IFN-γ in 5 of the 7 patients examined with N-specific CD4$^+$ T cells and 5 of the 6 patients examined with N-specific CD8$^+$ T cells (Fig. 3d, e). The variation in the co-expression was dramatic; in most patients, we detected less than 50% of IFN-γ$^+$ CD4$^+$ and CD8$^+$ T cells co-expressing TNF-α or GzmB (Fig. 3d, e). A similar varied and overall low proportion of co-expression was observed in convalescent patients with COVID-19 in the United Kingdom[23] and in Sweden[24].

**Virus-specific memory CD4$^+$ and CD8$^+$ T cells in recovered patients**. We then characterised the phenotypic memory of SARS-CoV-2-specific CD4$^+$ and CD8$^+$ T cells by CD45RA and CCR7 staining to determine the frequency of the naïve (CD45RA$^+$CCR7$^+$), central memory (CD45RA$^-$CCR7$^+$), effector memory (CD45RA$^-$CCR7$^-$), and late effector (CD45RA$^+$CCR7$^-$) subsets (Fig. 4a). In response to both rAd5-S and rAd5-N, the IFN-γ$^+$ CD4$^+$ T cells were phenotypically effector memory (CD45RA$^-$CCR7$^-$) and effector (CD45RA$^+$CCR7$^-$) cells (Fig. 4b). We observed similar constitutions for IFN-γ$^+$ CD8$^+$ T cells in response to rAd5-S and rAd5-N (Fig. 4b). However, in contrast, the virus-specific CD4$^+$ and CD8$^+$ memory T cells were highly heterogeneous (Fig. 4c). Additionally, only three patients had detectable memory CD8$^+$ T cells in response to rAd5-S. Of note, varied the frequencies of virus-specific CD4$^+$ and CD8$^+$ T cells we

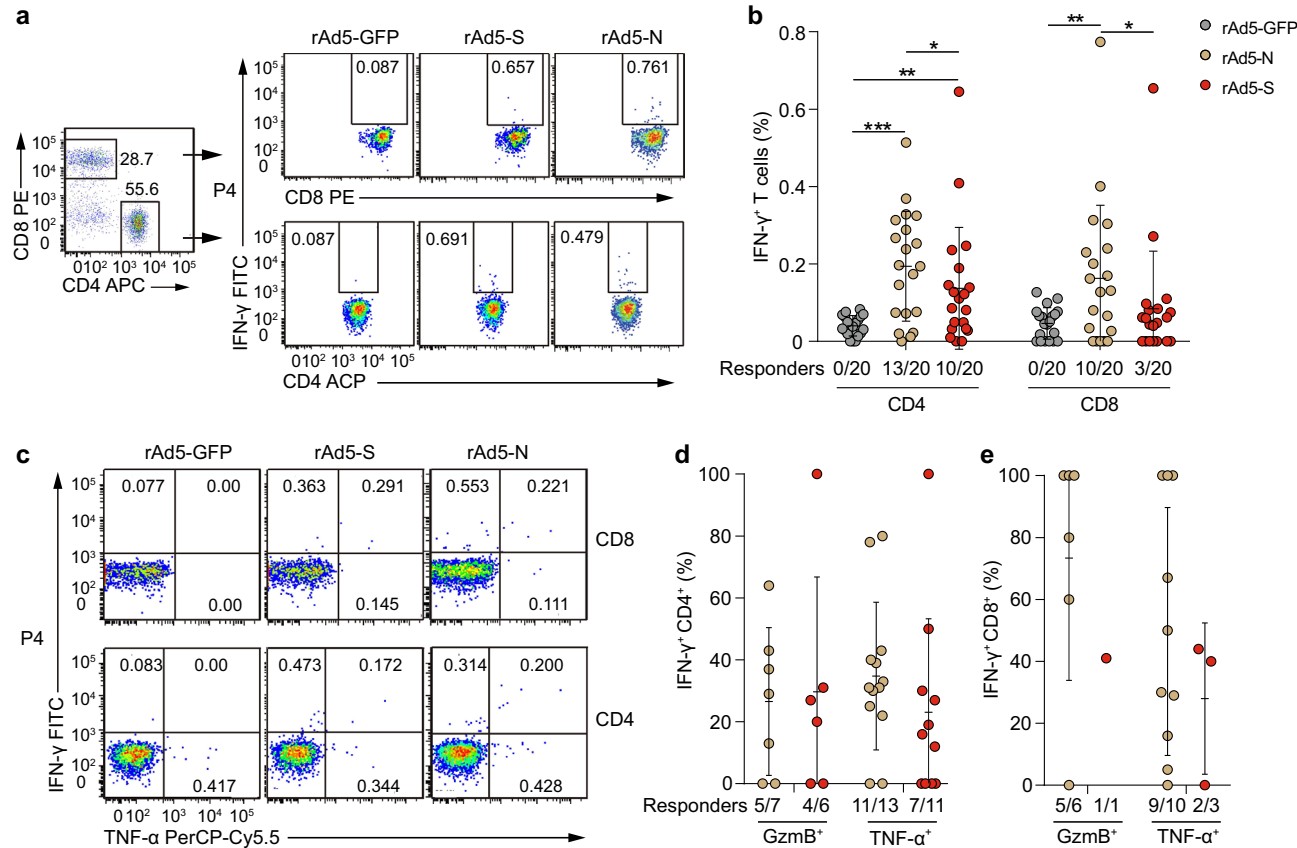

**Fig. 3 SARS-CoV-2-specific T cell responses in recovered COVID-19 patients 3–4 months after infection. a** Fluorescence-activated cell sorting (FACS) plot example for analysing IFN-γ expression in CD4 and CD8 T cells is shown for patient No. 4 (Pt4). **b** Percentage of CD4$^+$ and CD8$^+$ T cells producing IFN-γ in response to a recombinant replication-deficient adenovirus type 5 that encodes green fluorescent protein (rAd5-GFP), SARS-CoV-2 spike (rAd5-S), or SARS-CoV-2 nucleocapsid protein (rAd5-N) in PBMCs from recovered patients. **c** FACS plot examples of IFN-γ and granzyme B (GzmB) or TNF-a co-expression. **d–e** Functional profile of IFN-γ$^+$ CD4$^+$ (**d**) and CD8$^+$ (**e**) T cells producing GzmB, or TNF-a in response to rAd5-S and rAd5-N. Data are expressed as mean ± SD for **b**, **d** and **e**. *$p < 0.05$, **$p < 0.01$, and ***$p < 0.001$ by two-tailed paired t-test for **b** ($p = 0.0001$, $p = 0.0085$, and $p = 0.03$ for CD4, and $p = 0.009$ and $p = 0.174$ for CD8). The two-tailed Mann–Whitney U-test was used for **d** and **e**. Gating strategies are in Supplementary Fig. 5. Source data included as a Source Data File.

detected in patients in this study exhibited similar phenotypes, suggesting that infection produced poor T cell memory. Further assessment of the correlation between SARS-CoV-2-specific CD4$^+$ and CD8$^+$ T cell responses and antibody titre showed a moderate correlation between CD4$^+$ T cell responses and anti-N IgG antibody (Spearman $r = 0.53$, $p = 0.02$ for rAd5-N, and Spearman $r = 0.49$, $p = 0.039$ for rAd5-S) (Supplementary Fig. 1), anti-S IgG (Spearman $r = 0.53$, $p = 0.02$ for rAd5-N, and Spearman $r = 0.49$, $p = 0.039$ for rAd5-S) (Supplementary Fig. 2), and anti-RBD IgG (Spearman $r = 0.495$, $p = 0.031$ for rAd5-N, and Spearman $r = 0.486$, $p = 0.041$ for rAd5-S) (Supplementary Fig. 3), whereas we found no correlation between CD8$^+$ T cell responses and antibody titre or CD4$^+$ T cell response and NAb titre (Supplementary Figs. 1–4).

## Discussion

The need to understand the kinetics of antibody and T cell responses to SARS-CoV-2 is critical. This study prospectively evaluated the durability of the SARS-CoV-2-specific antibody and the T cell responses in patients 3–4 months after infection. We were able to detect anti-S, anti-RBD or anti-N IgG and NAb >14 days after infection in all patients during hospitalisation and observed no drastic decline in IgG and NAb levels 3–4 months after infection. In contrast, similar to other findings[25,26], we observed a rapid decline in anti-RBD IgM responses in the serum.

We also detected SARS-CoV-2-specific CD4$^+$ and CD8$^+$ T cell responses in approximately 65% of patients 3–4 months after infection. In summary, our data show durable IgG and NAb responses in all patients with COVID-19 and virus-specific T cell responses in most patients 3-4 months after infection.

Previous studies have shown that antibody responses against SARS-CoV and MERS-CoV infections can persist for at least 2 years[27–30]. Investigating antibody responses to SARS-CoV-2, recent studies have shown a robust increment of IgG and NAb levels in patients after 2–3 weeks PSO or in early convalescent patients[4–7,21]. However, the kinetics and durability of these antibody responses have rarely been reported. We report longitudinal antibody profiles in SARS-CoV-2 patients using serial blood samples (from day 1 to day 121 PSO). We observed that SARS-CoV-2-specific IgG, IgM, or NAb could be detected in some patients within the first week of illness onset. In particular, over 50% of patients became seropositive for NAb. Moreover, >80% of patients within two weeks PSO had detectable IgG (81.3%), IgM (87.5%), and NAb (93.8%) antibodies against SARS-CoV-2, and all patients became seropositive for antibodies within 3 weeks PSO with antibody titre peaking around the same time. Notably, our findings that IgG and NAb against SARS-CoV-2 are relatively stable 3–4 months after infection are consistent with the two most recent studies, which likewise noted durability in the IgG response to the spike trimer or NAb response to

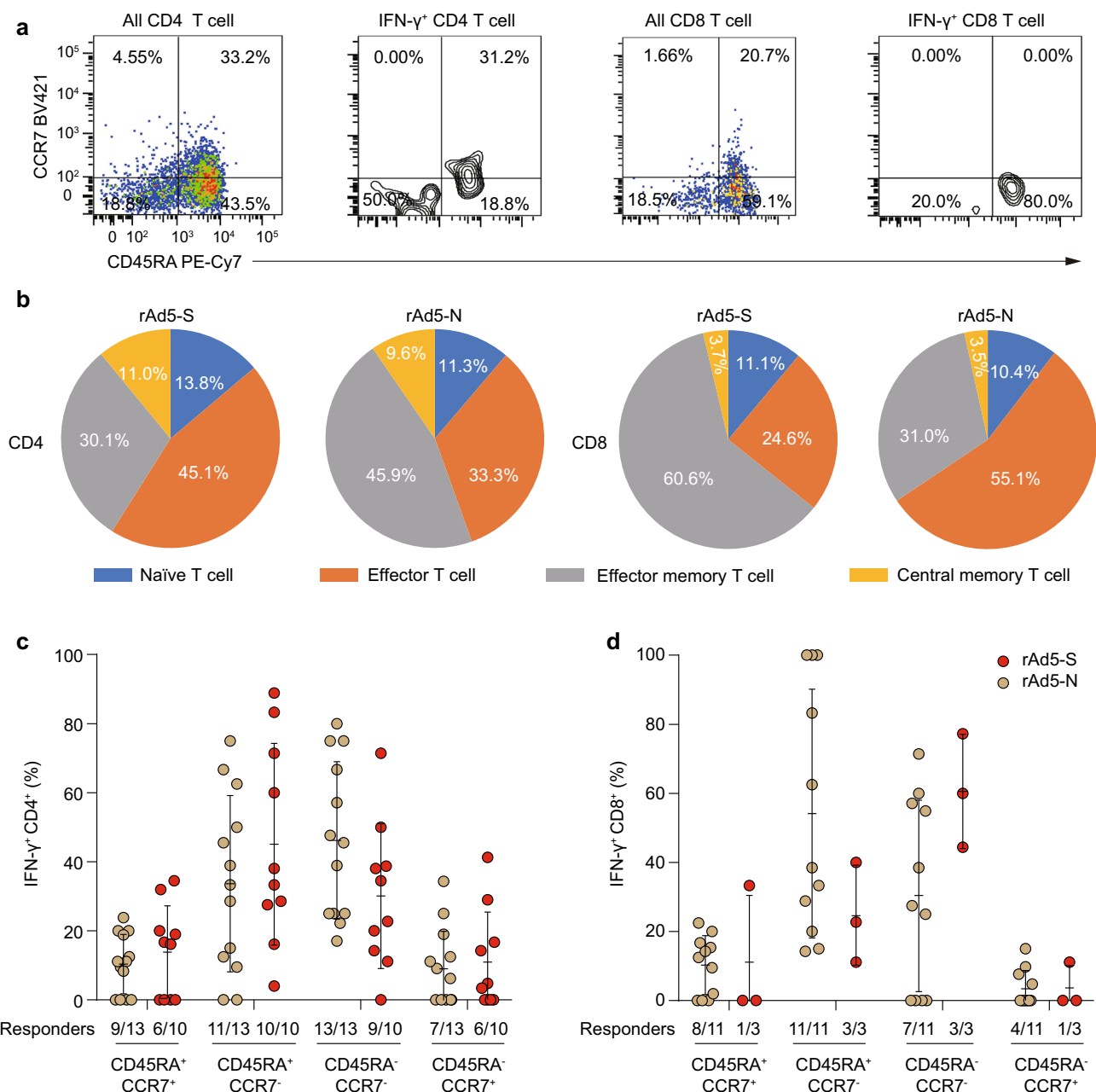

**Fig. 4 Phenotypic memory of SARS-CoV-2-specific T cells in recovered COVID-19 patients 3–4 months after infection.** Phenotypic memory (naive, CD45RA[+] CCR7[+]; central memory, CD45RA[−] CCR7[+]; effector memory, CD45RA[−] CCR7[−]; and late effector, CD45RA[+] CCR7[−]) analysis of IFN-γ-secreting CD4[+] and CD8[+] T cells. **a** FACS plot examples of CD45RA and CCR7 expression on all CD4 and CD8 T cells and recombinant replication-deficient adenovirus type 5 that encodes nucleocapsid protein (rAd5-N) -specific IFN-γ-secreting CD4[+] and CD8[+] T cells. **b** The constitution ratio of naive, central memory, effector memory, and late effector T cells on virus-specific IFN-γ-secreting CD4[+] and CD8[+] T cells of recovered patients 3–4 months after infection. Data on rAd5 that encodes green fluorescent protein (rAd5-GFP) controls are not shown because few virus-specific cytokine-secreting T cells were detected. **c–d** The percentage of rAd5-S- and rAd5-N-specific memory T cells in CD4[+] (**c**) and CD8[+] (**d**) T cells in recovered patients 3–4 months after infection. Data are presented as mean ± SD. The two-tailed Mann–Whitney *U*-test was used for **c** and **d**. Source data included as a Source Data File.

pseudovirus[25,26]. These data and ours contrast with those of Long et al.[8] and Ibarrondo et al.[31] showing a rapid decay of antibody levels. However, the factors contributing to the discrepant results are not entirely clear. In this study, in addition to employ ELISA used in other studies for IgG antibody detection[25,26], we also used a live-virus neutralising assay—a "gold standard assay"—to detect and measure NAb levels. Collectively, our results indicate that IgG and NAb persist at stable and high levels in the majority of patients 3–4 months after infection, which has important implications for vaccine development.

While it is essential to characterise the SARS-CoV-2-specific antibody response, it is also important to determine the SARS-CoV-2-specific T cell responses in recovered patients as memory T cells are known to protect against various viral infections[32]. Recent studies carried out in Australia, Germany, Sweden, the United Kingdom, and the United States have reported a particularly high frequency of S-specific CD4[+] T cell responses among COVID-19 early convalescent patients approximately 1 month PSO[7,21,23,24,33–35]. Habel et al. reported suboptimal S-specific CD8[+] T cell responses associated with the prominent HLA-

A*02:01 phenotype in Australian Caucasian COVID-19 convalescent patients[35]; other studies have detected robust S-specific CD8[+] T cell responses[21,23,24]. In our study, we observed S-specific CD4[+] and CD8[+] T cell responses in 50% and 15% of the recovered Chinese COVID-19 patients, respectively. However, it is impossible to rule out, as potential confounding factors causing for such differences, the technical limitation of the assays used in our and other studies. Moreover, HLA genotypes may have played an important role. Another possibility is that S-specific T cells, especially for CD8[+] T cell responses, degraded over the 3–4 months period.

On the other hand, the reported findings of SARS-CoV-2 N-specific T cell responses are highly controversial. Grifoni et al. reported that N protein contributes only about 10% to the total CD4[+] and CD8[+] T cell responses among convalescent patients in the United States[21]. Peng et al. found that the overall N-specific T cell responses were much lower than the S-specific responses, but found a higher proportion of multifunctional M/N-specific CD8[+] T cells compared with S-specific T cells among patients who had recently just recovered from mild illness in the United Kingdom[23]. However, Habel et al. similarly reported robust CD4[+] T cell responses but weak CD8[+] T cell responses directed against N and S proteins among Caucasian COVID-19 convalescent patients in Australia[35]. Le Bert et al. detected CD4[+] and CD8[+] T cells that recognised multiple regions of the N protein among all convalescing patients tested in Singapore[36]. Furthermore, among COVID-19 convalescent patients in Hong Kong, Zhou et al. reported higher N-specific CD4[+] and CD8[+] T cell responses than RBD-specific responses[37]. Sekine et al. similarly reported robust CD4[+] and CD8[+] T cell responses directed against N and S proteins among convalescing individuals with asymptomatic and mild COVID-19 in Sweden[24]. In our study, we observed N-specific CD4[+] and CD8[+] T cell responses in 65% and 50%, respectively, of recovered Chinese COVID-19 patients, respectively. Thus, N-specific T cell responses, especially for CD8[+] T cells, are more robust than S-specific responses. Several factors may contribute to differences across studies. First, Grifoni et al. used predicted epitopes that capture about 50% of the total CD4[+] T cell responses and target the 12 most prominent HLA class I A and B alleles[21]. In this way, they may have narrowed their peptide patterns, and missed some epitopes. In contrast, we and others employed the entire N protein or overlapping peptides that covered the whole N protein. Second, the HLA genotypes have affected the responses. Hebel et al. chose the HLA-A*02:01 phenotype and detected suboptimal CD8[+] T cell responses[35]. Peng et al. identified SARS-CoV-2 CD8 optimum epitopes restricted by B*2705, B*0702, B*4001, A*0301, A*1101, and A*0101[23]. Third, the experimental conditions, such as geographical and temporal variations, may also have contributed to the differences. Together, these results suggest the potential importance of including non-spike proteins within future COVID-19 vaccine design.

Previous studies have also reported the two main phenotypic memory T cells as effector memory (CD45RA[−]CCR7[−]) and central memory (CD45RA[−]CCR7[+]) CD4[+] T cells in recovered SARS and MERS patients[38–40], and the persistence in circulation of the late effector (CD45RA[+]CCR7[−]) CD8[+] T cells. Each subset of these T cells plays a role in the protective immunity to reinfection by rapidly migrating effector subsets into tissues to provide protection and proliferating central memory T cells in the draining lymph node, so providing a pool of new effector cells[41]. Peng et al.[23] reported SARS-CoV-2-specific CD8[+] T cells among convalescent patients in the United Kingdom were mainly effector memory (CD45RA[−]CCR7[−], 50.3% ± 13.3%) and central memory (CD45RA[−]CCR7[+], 20.7% ± 8.4%) phenotypes. Zhou et al. found similar trends for both CD4[+] and CD8[+] T cells

responsive to SARS-CoV-2 N protein and RBD among COVID-19 convalescent patients in Hong Kong[37]. In our study, although we detected varied frequencies of SARS-CoV-2-specific CD4[+] and CD8[+] T cells in patients, the majority of S- and N-specific CD4[+] and CD8[+] T cells were phenotypically effector memory (CD45RA[−]CCR7[−]) and late effector (CD45RA[+]CCR7[−]) T cells. This phenomenon was also observed in COVID-19 patients about 1 month after infection[34]. It is possible that S- and N-specific T cells expressing the central memory (CD45RA[−]CCR7[+]) phenotype fall off rapidly after the infection has resolved.

Our study had several limitations. The small sample size, especially for severe patients and asymptomatic individuals, was dictated limited by expediency, which limited the analysis of the antibody responses stratified by patient age, sex, underlying condition, or disease severity. In addition, our follow-up of patients is currently at 3–4 months post-infection. Thus, assessment of the duration and resiliency of the SARS-CoV-2 antibody and T cell responses in a large cohort study would be desirable for validation of our results. Last, taking into consideration the biosecurity issue, because of the unavailability of instrumentation in the BSL-3 facility as SARS-CoV-2 RNA has been detected in patient's blood samples[42], we did not isolate serial PMBCs from patients during their hospitalisation which also limited to full characterisation of the dynamics of the T cell responses during infection.

In summary, we measured the dynamics of SARS-CoV-2-specific antibodies and CD4[+] and CD8[+] T cell responses in COVID-19 patients. All patients not lost to follow-up had high levels of antibodies 3–4 months after infection. We detected SARS-CoV-2-specific CD4[+] and CD8[+] T cells in approximately 65% of recovered patients. Our findings can inform the design of future serological studies and the development of SARS-CoV-2-targeted vaccines.

## Methods

**Ethics statement**. All patients provided written informed consent. The study was conducted following the Declaration of Helsinki, and the Institutional Review Board of the Academy of Military Medical Sciences approved the study protocol (IRB number: AF/SC-08/02.60).

**Study design and participants**. From January 12, 2020, through February 14, 2020, we invited patients hospitalised with SARS-CoV-2 infection in the local hospitals in Linyi City of Shandong Province, China, to give informed consent to participate in this study. All potential patients had a diagnosis of SARS-CoV-2 confirmed by positive real-time reverse transcription-polymerase chain reaction (RT-PCR) results. We enrolled a total of 16 newly diagnosed patients, and prospectively collected their blood samples up to 3–4 months after illness onset. Of these 16 patients, 5 were lost to follow up at 3–4 months after infection. At the 3–4 month follow up visit, we additionally enrolled 9 recovered patients 3–4 months after infection. We collected blood from each participant using serum separator tubes with gel during hospitalisation and at hospital discharge. At the 3-4 month follow up visit, we collected blood was collected in an EDTA tube and serum separator tube for PBMCs and serum isolation, respectively. To compare the proportion of patients with detectable antibody responses at different time points post-symptom onset (PSO), we assessed serum samples collected ≤7 days (median 5, interquartile range [IQR] 3–6), ≤14 days (median 10, IQR 9–12), and ≤21 days (median 17, IQR 15–19) PSO; during 22–42 days (median 28, IQR 24–30) after onset; and then at 3-4 months (median 106, IQR 103–109). We recorded the demographic and clinical characteristics of patients, including baseline demographic data, date of symptoms onset, presenting symptom including fever, cough, sputum production, and sore throat, past medical and smoking history, hospitalisation, and radiological evidence of complication by pneumonia, were collected at enrolment. We additionally used 10 age- and sex-matched healthy control subjects whose serum samples had been collected before the pandemic as controls.

**Case and disease severity definition**. We defined a laboratory-confirmed patient of COVID-19 as an individual positive for SARS-CoV-2 by RT-PCR of naso-pharyngeal swabs. We defined a symptomatic patient as an individual with laboratory-confirmed COVID-19 with symptoms such as fever, cough, sore throat, sputum, and so on. We defined a patient with an asymptomatic infection as an individual who was positive for SARS-CoV-2 by RT-PCR without any relevant

symptoms. According to the diagnostic and treatment guideline for SARS-CoV-2 issued by the Chinese National Health Committee (Version 7), we defined mild illness as having mile clinical symptoms but without radiological signs of pneumonia; we defined moderate illness according to the following criteria: (i) fever and respiratory symptoms and (ii) radiological signs of pneumonia; we defined severe illness as satisfying at least one of the following items: (i) breathing rate ≥30/min; (ii) pulse oximeter oxygen saturation (SpO$_2$) ≤93% at rest; (iii) ratio of the partial pressure of arterial oxygen (PaO$_2$) to a fraction of inspired oxygen (FiO$_2$) ≤300 mm/Hg (1 mm/Hg = 0.133 kPa).

**Serum and PMBCs isolation**. We collected venous blood from each participant to separate serum or isolate PBMCs. We separated sera by centrifugation at 800 $g$ for 10 min, aliquoted into three cryovials, and preserved at −80 °C until testing. We isolated PBMCs by density-gradient sedimentation using Lymphoprep™ density gradients (Axis-Shield, Norway). Isolated PBMCs were frozen in cell recovery media containing 10% DMSO (GIBCO), supplemented with 90% heat-inactivated foetal bovine serum (FBS), and stored them in liquid nitrogen before assay analyses.

**Qualitative SARS-CoV-2 IgG and IgM detection**. To measure serum IgG to nucleoprotein (N) and IgM to the RBD of the spike (S) protein of SARS-CoV-2, we performed ELISAs using two well-validated commercial diagnostic ELISA kits (Beijing Wantai Biological Pharmacy Enterprise Co., Ltd)[4]. We detected the IgG antibodies using an indirect ELISA kit based on a recombinant nucleoprotein of SARS-CoV-2. Briefly, a 100 µl dilution buffer was added into the wells except for three positive wells, three negative wells, and one blank. We then added 100 µl of positive control, of negative control and 10 µl of serum specimen into their respective wells (except for the blank well) and incubated them at 37 °C for 30 min. After washing each well 5 times with diluted wash buffer, 100 µl of HRP-conjugated goat anti-human IgG was added into each well apart from the blank well, and incubated them at 37 °C for 30 min. After washing each well 5 times, 50 µl of chromogen solution A and then 50 µl of chromogen solution B were added into each well and incubated at 37 °C for 15 min, avoiding light. The reaction was visualised by adding 50 µl of stop solution into each well. A spectrophotometer measured the optical density (OD) at 450 nm and 630 nm. We used the IgM µ-chain capture ELISA to detect IgM antibodies, where mammalian cell-expressed recombinant antigens contained RBD of the S protein of SARS-CoV-2 as the immobilised and HRP-conjugating antigen. Briefly, the detection procedure was consistent with IgG detection apart from our not adding dilution buffer to the blank well, using the same volume of 10 µl for positive control, negative control and serum specimen, and HRP-conjugating RBD antigen. The cut-off value for IgG is the mean OD value of three negative controls (if the mean absorbance value for three negative calibrators is < 0.03, take it as 0.03) + 0.16, whereas the cut-off for IgM is the mean OD value of three negative controls (if the mean absorbance value for three negative calibrators is < 0.03, take it as 0.03) × 2.1. A serum sample with an OD value ≥cut-off OD value was considered to be an anti-N IgG or anti-RBD IgM antibody positive.

**ELISA analysis of serum IgG antibody to RBD and spike trimer**. To further quantify the serum IgG antibody response to RBD and S of SARS-CoV-2[12], the recombinant RBD and S trimer derived from SARS-CoV-2 (Sino Biological, Beijing) were coated onto flat-bottom 96-well plates overnight at 4 °C with a final concentration of 1 µg/ml. Plates were washed with PBS-T (PBS with 0.05% Tween 20) and blocked with blocking buffer (5% skim milk and 2% BSA in PBS) for 1 h at room temperature. Duplicate 3-fold 8-point serial dilutions (starting at 1:100) of heat-inactivated serum samples diluted in 1% milk in PBS-T were added to the wells and incubated at 37 °C for 1 h. Wells were then incubated with secondary anti-human IgG antibody labelled with HRP (Promega, W4031, 1:5000 Promega,) and TMB substrate (Kinghawk, Beijing). The optical density (OD) was measured by a spectrophotometer at 450 nm and 630 nm. Endpoint antibody titre was calculated as the reciprocal serum dilution giving signal three times that of the healthy controls using a serum titration starting at 1:100 and using a 3-fold dilution series by a fitted curve (4 parameter log regression).

**Serum neutralising antibody against live SARS-CoV-2**. We measured the serum neutralising antibody by a live-virus neutralising assay. Briefly, serum samples were heat-inactivated at 56 °C for 30 min and diluted from 1:40 with a serial two-fold dilution in microtiter plates. Then 50 µl of previously titrated tissue culture infecting dose 50 (TCID$_{50}$) of SARS-CoV-2 (BetaCoV/Wuhan/AMMS01/2020) was added to each serum dilution in duplicate. After a 1 h incubation at 37 °C, 5% CO$_2$, 50 µl of 1×10$^4$ Vero E6 cells were added to each well of the virus-serum mixture. The mixture was incubated in 96-well plates for 2 days (48 h), after which the cytopathic effect (CPE) was read. Serum neutralising antibody titer was defined as the reciprocal of the highest dilution showing a 100% CPE reduction compared to the virus control. Virus-only controls and cell-only controls were included in each neutralisation assay plate. For final titre <40, we assigned a value of 20 for geometric mean calculations and was considered seronegative.

**Flow cytometry**

*Virus-specific T cells stimulation*. PBMCs were stimulated with recombinant replication-deficient adenovirus type 5 vector encoding green fluorescent protein (rAd5-GFP, as control), SARS-CoV-2 spike (rAd5-S), or SARS-CoV-2 nucleocapsid (rAd5-N) at MOI 100 (GenBank: MN908947.3, Vigenebio, Jinan, China) in plain RPMI-1640 for 1 h at 37 °C. Then 10% FBS and 20 U/ml human interleukin-2 (IL-2, R &D Systems, Abingdon, United Kingdom) were added to the cultures. After 66 h incubation, 1 µl/ml brefeldin A (BFA, Cat No. 00-4506-51, eBioscience, San Diego, CA, USA) was added to block cytokine secretion for 6 h, whereafter, the cells were subjected to flow cytometry analysis.

*Surface staining of surface markers and cytokines*. For surface staining, single-cell suspensions were washed once with fluorescence-activated cell sorting (FACS) washing buffer (2% FBS, 0.1% NaN3 in PBS) and blocked with Fc receptor blocking solution. Cells were then incubated with fluorescence-conjugated antibodies against cell surface molecules for 30 min at 4 °C. After washing with FACS buffer, the cells were fixed and permeabilized using a fixation/permeabilisation kit (eBioscience) and stained with fluorescence-conjugated specific antibodies against cytokines following the manufacturer's instructions. Flow cytometry was performed using a Becton Dickinson FACS Canto (BD Biosciences, San Jose, CA, USA), and the data were analysed with the FlowJo software (Flowjo, Ashland, OR, USA). The following anti-human monoclonal antibodies were used in the staining assay: BV510-labelled anti-CD3 (300448), APC-labelled anti-CD4 (357408), PE-labelled anti-CD8a (300908), PE-Cy7-labelled anti-CD45RA (304126), BV421-labelled anti-CCR7 (353208), PerCP-Cy5.5-labelled anti-TNF-α (502926), BV421-labelled anti-GzmB (396414), and FITC-labelled anti-IFN-γ (11-7319-82); all antibodies were from eBioscience or BioLegend.

**Statistical analysis**. We analysed the anti-S and anti-RBD IgG and NAb titre with log$_{10}$-transformed geometric means and 95% confidence intervals (95%CI), and determined the mean and 95%CI of the OD value for anti-N IgG and anti-RBD IgM. The log-transformed mean and 95%CI were then back-transformed to the original scale. We calculated the proportion of antibody titre equal to or greater than the threshold and associated 95%CIs. We used the two-tailed paired $t$-test or two-tailed Mann–Whitney $U$-test for testing the differences in virus-specific T cell responses. We used nonparametric Spearman correlation analyses to determine associations between analysed parameters. All statistical tests were 2-sided with a significance level of 0.05. We performed all statistical analyses in Prism (GraphPad Software).

**Reporting summary**. Further information on research design is available in the Nature Research Reporting Summary linked to this article.

## Data availability
The data that support the findings of this study are available within the paper and its supplementary information files. Source data are provided with this paper.

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

## Acknowledgements

We thank all patients for their participation in this study. This work was supported by grants from the Natural Science Foundation of China (81773494 to M.-J.M.), the Beijing Natural Science Foundation (L202038 to M.-J.M.), the National Major Project for Control and Prevention of Infectious Disease of China (2017ZX10303401-006 to M.-J.M.), the National Key Research and Development Program of China (2016YFC1303402 to J.-Y. Z, 2020YFC0846100 to Y.-W. G.), and the Key Research and Development Program of Shandong Province (2020SFXGFY02 to D.-M. K).

## Author contributions

M.J.M. conceived the study. X.L.J., S.X.J., X.L.Z., C.B.L., Z.Q.K., Y.L., Y.W.Z., Q.D., and D.M.K. collected clinical samples; G.L.W., X.N.Z., X.W., L.J.D., and L.Y. performed serological testing; Y.W.G., F.H.Y., T.C.W., E.T.L., X.F.W., and W.Y.S. performed neutralising assay at a P3 laboratory. J.Y.Z. and Q.Y.W. performed flow cytometry assay; M.J.M., G.L.W., L.J.D., L.Y., J.Y.Z. analysed the data; and M.J.M. and J.Y.Z. drafted the manuscript. All authors reviewed and approved the final manuscript.

## Competing interests

The authors declare no competing interests.
