## [Peer Review File · Nature Communications]

REVIEWER COMMENTS

Reviewer #1 (Viral immunity, antibody responses) (Remarks to the Author):

The half-life of the neutralizing antibody response to SARS-CoV2 will substantially impact whether and how it will be possible to achieve global herd-immunity. The manuscript by Jiang et al contributes some answers to this important question and challenges a previous report (Long et al., 2020) claiming that SARS-CoV2 neutralizing titers wash out rapidly.

The authors assembled a small cohort of SARS-CoV2 infected individual and assessed several immune parameters at different time-points following onset of symptoms. The humoral immunity parameters assessed include IgG antibodies to the nucleoprotein, IgM antibodies to the receptor binding domain (RBD) and neutralizing titers. However, there are several major issues with the way these parameters were measured. First, from the manuscript the exact parameters parameters that were measured in Figure 1 are difficult to deduct. The corresponding method section is ambiguous, but it has to be assumed that IgM titers were assessed against the RBD domain while IgG titers were against measured against the nucleoprotein. Also, ELISA results were presented as OD405 from a single serum dilution that was not specified, no details were provided as to how different assays were standardized to each other, and how potential issues with substrate saturation were prevented. Last, control samples from healthy individuals are missing. The use of different antigens to detect IgG or IgM antibodies, drastically limits the comparability of the results and only little immunological information is gained from their comparison. It would have made considerably more sense to measure anti-RBD and anti-S IgG instead of nucleoprotein.

Next, the authors assessed the T cell responses to SARS-CoV2, and again, their findings are only partially in line with the previously published studies [e.g. to (Grifoni et al., 2020)] with little discussion or interpretation to explain possible reasons contributing to these differences.

The manuscript lacks experimental details in essential sections, and the study suffers from random study design decisions (e.g. N vs S titers). This impairs the usability of the data presented here, and makes it difficult put the presented results in context with other work. The overall writing is difficult to follow and much of the methodology is omitted, limiting a reader's ability to follow any arguments or judge the strength of conclusions based on any given experiment.

As such, the primary value of this study lays in the observation that neutralizing antibody titers did not wash out as quickly as previously assumed. However, as with the their findings regarding T cell responses, the observed differences to other studies are not satisfactorily explained or discussed. The manuscript cannot be accepted in its current form and needs major revisions before it could be reconsidered.

General issues

Manuscript:

The manuscript would benefit from proof reading to smooth out the use of English, several sections are quite difficult to follow because of word usage. Also, it is written in a manner that assumes a high level of technical familiarity with various abbreviations, techniques, immune cell definitions (i.e. phenotypically effector memory (CD45RA⁻ CCR7⁻) and effector (CD45RA⁺CCR7⁻) cells in response to both rAd5-S and rAd5-N rather than effector memory and central memory T cells" (no explanation of how the effector memory and central memory cells would differ from what they observe), not suitable for wide readership in Nat Comm.

Methods:

Do not clearly explain their methodologies in the results section, meaning one has to hope it is explained in the figure captions or methods and then piece together the information

Figures:

Figures are overloaded and text labels are often too small. The use of larger font-sizes would be advisable.

Sup. Data:

Weak supporting data, one experimental setup for each claim, no considerations of alternate explanations or limitations of their experimental systems

Minor issues

Line 29: Clarify the wording of IgG and IgM, assume reference to SARS2-spike binding IgG & IgM? Or is it whole IgG & IgM in the serum?

L55-58: referencing is inconsistent; refs 7,8,9 should also be referenced in the statement in line 55.

Line 63: meaning not clear; perhaps 'or not present at all'?

Line 82: 'the same ... RBD antigen as the Ab-ELISA'? What is Ab-ELISA referring to?

Line 93: how were IgG and IgM measured? Against what antigen? Not stated in the text.

Line 95: It is not clear which antigen the titers were assessed. From the method section it could be both anti-nucleoprotein or anti-RBD antibodies. Please clearly indicate the parameters assessed here, including the serum dilutions used.

Line 109: clarify that NAb was detected out of whole sera

Line 110: define acronym GMT? What units does GMT have?

Line 120: which Ig correlates better with Neut? IgG or IgM?

Line 122: was IgG only measured against N (define acronym) and IgM against S (define acronym)?

Can the authors provide supporting evidence that the IgG does not bind to S, or that IgM doesn't bind to N? It seems unlikely there would be an isotype specific antigen recognition related to variable domains of antibodies.

Line 131: are control healthy donors also tested with the Ad5 recognition assay?

Line 134: the authors measure T cell responses 3-4 months after recovery, yet claim based on the behaviour of these post-infection T cells that T cell function was impaired during infection. Can the authors support this statement with any further data? Can they, for instance, be certain that the S and N specific T cell response has not degraded over the 3-4 months after active infection?

Line 167: Why unexpectedly?

Line 181: 'Of interesting, in comparison with negative IgG detection in 12.9% of symptomatic patients 8 weeks after discharge, ...'. Meaning 'Interestingly, compared to 12.9% negative IgG levels published for symptomatic patients 8 weeks following discharge, ..'?

Line 191: English incorrect

Line 194: Duplication of the word 'However'

Line 205: Not clear what authors mean.

Line 277ff: It is not clear what was measured.

Figure 1: It is not clear whether the IgM and IgG titers indicated are against the nucleoprotein or the RBD. Also, how are Nab titers presented in these graphs? What do the 1, 2, 3, 4 mean?

Figure 2: Why did the authors use SEM instead of SD? Is a change of IFG-g+ between 0.04% and 0.2% meaningful? How does the magnitude of the measures for these experiments compare with other work? How does it compare to healthy donors, do they react to any of the Ad5s?

Ext. Fig.1: The figure is full of white space, yet the information is printed so small that it barely can be read. Range of axes need to be adjusted to reflect the data represented and label and symbol sizes need to be increased.

Ext. Fig.2: Too low resolution bitmaps with percentages so small they cannot be read on a printout. It

appears that the singlet gating on FSC-W/FSC-H and SSC-W/SSC-H graphs are not visible. Is the upper IFN-g graph from CD8+ and the lower from CD4+? They are not labelled.

Reviewer #2 (Vaccination, neutralizing antibody) (Remarks to the Author):

This study characterized serum antibody binding and neutralizing responses, and CD4 and CD8 T cell responses, to SARS-CoV-2 infection out to 3-4 months post-infection in a cohort of 25 patients. The median age in the cohort was 40 years, roughly half were male (13/25), and most (18/25) of the cohort had moderate infection. To this reviewer, the most important findings are that 100% of the cohort produced binding and neutralizing antibodies by 14 days post-infection (and >80% produced such responses by day 10); the neutralizing titers were high in a live-virus neut assay at 15-21 days post-infection (GMT = 1290; 95% CI 873-1875); and the neutralizing titers were relatively sustained out to three months post-infection, decaying by less than a factor of two (GMT = 697; 95% CI 401-1215). The manuscript also found that up to 65% of patients mounted detectable virus-specific CD4 or CD8 memory responses at the 3-4 month timepoint.

The relatively high durability of the neutralizing responses detected here are an important early indication that infection potentially might induce relatively well-sustained protective antibody responses, and these data also provide at least one important benchmark for ongoing vaccine studies. This reviewer believes that these are timely and important findings for the field to consider.

minor issues:

The section in lines 54-68 fails to cite or discuss many papers published characterizing neutralizing antibody responses to SARS-CoV-2. For example: Brouwer, van Gils et al Science 2020; Cao, Xie et al Cell 2020; Hansen, Kyrtatsous et al Science 2020; Ju, Zhang et al Nature 2020; Kreer, Klein et al Cell 2020; Robbiani, Nussenzweig et al Nature 2020; Rogers, Burton et al Science 2020; Seydoux et al Immunity 2020; Wec, Walker et al Science 2020; Wu, Liu et al Science 2020.

lines 135-137 has a typo and conflicts with lines 190-192. lines 135-137 say "approximately 65% of recovered patients could not produce SARS-CoV-2 specific CD4 or CD8 T cell response 3-4 months after infection", while lines 190-192 say "We found that only approximately 65% of patients had detected SARS-CoV-2-specific CD4+ or CD8+ T cells 3-4 months after infection."

The CD4 and CD8 response rate (% responders) is not clearly indicated in Fig. 2, although the authors give such response rates in the text. It would be helpful to add information or a panel to this figure giving the response rates, similar to the "positive rate" shown in Fig 1A,B for antibody responses.

It is unclear to this reviewer if the detection of CD4 or CD8 responses responses in 65% or less of patients might or might not be due to technical limitations of the assay used. Were prior analyses of cellular responses to SARS-CoV-2 that showed higher response rates at 1 month post-infection carried out using a similar or different assay?

Response to reviewers' comments

Reviewer 1

(1) The half-life of the neutralizing antibody response to SARS-CoV2 will substantially impact whether and how it will be possible to achieve global herd-immunity. The manuscript by Jiang et al contributes some answers to this important question and challenges a previous report (Long et al., 2020) claiming that SARS-CoV2 neutralizing titers wash out rapidly.

[Response] We thank the reviewer for this positive assessment of our study.

(2) The authors assembled a small cohort of SARS-CoV2 infected individual and assessed several immune parameters at different time-points following onset of symptoms. The humoral immunity parameters assessed include IgG antibodies to the nucleoprotein, IgM antibodies to the receptor binding domain (RBD) and neutralizing titers. However, there are several major issues with the way these parameters were measured. First, from the manuscript the exact parameters that were measured in Figure 1 are difficult to deduct. The corresponding method section is ambiguous, but it has to be assumed that IgM titers were assessed against the RBD domain while IgG titers were against measured against the nucleoprotein. Also, ELISA results were presented as OD405 from a single serum dilution that was not specified, no details were provided as to how different assays were standardized to each other, and how potential issues with substrate saturation were prevented. Last, control samples from healthy individuals are missing. The use of different antigens to detect IgG or IgM antibodies, drastically limits the comparability of the results and only little immunological information is gained from their comparison. It would have made considerably more sense to measure anti-RBD and anti-S IgG instead of nucleoprotein.

[Response] The reviewer raises an important point, and we agree that our study would benefit from detailed information on methods and evaluation of anti-RBD and anti-S IgG antibody. We have added more detailed information for the detection of IgG and IgM antibodies in the Methods section of the revised manuscript (Line 418-445). Second, the two ELISA kits for detecting IgG and IgM antibodies were commercial kits and have been well-validated for diagnosing COVID-19. However, these two ELISA kits are intended to detect IgG and IgM qualitatively. Therefore, only a single serum dilution (1:10) was conducted per the manufacturer's instructions. More details have been added in the Methods section of the revised manuscript (Line 418-445). Third, we have added information for healthy controls in revised Figure 1 and described this in the main text (Line 126-128, 138-139, 156-157). Fourth, we additionally conducted ELISA to quantify the anti-RBD and anti-S IgG antibody. The methods and results data have also added in the Methods section (Line 447-460) and the Results section (Line 129-139) in the revised manuscript.

(3) Next, the authors assessed the T cell responses to SARS-CoV2, and again, their findings are only partially in line with the previously published studies [e.g. to (Grifoni et al., 2020)] with little discussion or interpretation to explain possible reasons contributing to these differences.

[Response] Thanks! We have added more discussion in the Discussion section of the revised manuscript (Line 282-322) as follows: “...a particularly high frequency of S-specific CD4⁺ cell responses was observed in recent studies by several independent groups in the USA, Australian, UK, Sweden, and Germany COVID-19 early convalescent patients approximately 1 month from symptom onset (Braun et al., 2020; Grifoni et al., 2020; Habel et al., 2020; Ni et al., 2020; Peng et al., 2020; Sekine et al., 2020; Weiskopf et al., 2020). Habel et al. reported suboptimal S-specific CD8⁺ T cell responses associated with the prominent HLA-A*02:01 phenotype in Australian Caucasian COVID-19 convalescent patients (Habel et al., 2020); others have detected robust S-specific CD8⁺ T cell responses (Grifoni et al., 2020; Peng et al., 2020; Sekine et al., 2020). In our study, S-specific CD4⁺ and CD8⁺ T cell responses were observed in 50% and 15% of the recovered Chinese COVID-19 patients, respectively. However, the potential factors for such difference cannot rule out due to the technical limitation of the assays used because we don't have enough controls. Moreover, it is possible that HLA genotypes may make differences. Another potential possibility is that S-specific T cells, especially for CD8⁺ T cell response, have degraded over the 3-4 months after infection.

On the other hand, the reported findings of SARS-CoV-2 N-specific T cell responses are highly controversial. Grifoni et al. reported that N protein only contributes about 10% to the total CD4⁺ and CD8⁺ T cell responses in USA convalescent patients (Grifoni et al., 2020). Peng et al. found that the overall N-specific T cell responses were much lower than S-specific responses, but there was a higher proportion of multifunctional M/NP-specific CD8⁺ T cells compared with S-specific T cells in UK patients who just recovered from mild illness (Peng et al., 2020). However, Habel et al. reported similarly robust CD4⁺ T cell responses but weak CD8⁺ T cell responses directed against N and S proteins in Australian Caucasian COVID-19 convalescent patients (Habel et al., 2020). Le Bert et al. detected CD4⁺ and CD8⁺ T cells that recognized multiple regions of the N protein in all the Singapore convalescing patients tested (Le Bert et al., 2020). Furthermore, Zhou et al. reported higher N-specific CD4⁺ and CD8⁺ T cell responses than RBD-specific responses in Hong Kong COVID-19 convalescent patients (Zhou et al., 2020). Sekine et al. reported similarly robust CD4⁺ and CD8⁺ T cell responses directed against N and S proteins in the Sweden convalescent individuals with asymptomatic and mild COVID-19 (Sekine et al., 2020). In our study, N-specific CD4⁺ and CD8⁺ T cell responses were observed in 65% and 50% of the recovered Chinese COVID-19 patients, respectively. Thus, N-specific T cell responses, especially for the CD8⁺ T cell responses, are more robust than S-specific responses. Several factors may contribute to differences across studies. First, Grifoni et al. used predicted epitopes that capture about 50% of the total CD4 T cell responses and target the 12 most prominent HLA class I A and B alleles (Grifoni et al., 2020). In this way, their peptide patterns may be narrowed, and some epitopes may be missed. In contrast, we and others employed whole N protein or overlapping peptides that covered the whole N protein. Second, the HLA genotypes may affect the responses. Habel et al. chose HLA-A*02:01 phenotype and detected suboptimal CD8⁺ T cell responses (Habel et al., 2020). Peng et al. identified SARS-CoV-2 CD8 optimum epitopes restricted by B*2705, B*0702, B*4001, A*0301, A*1101, and A*0101 (Peng et al., 2020). Third, the experimental

conditions, such as geographical and temporal variations, may also contribute to the differences.”

(4) The manuscript lacks experimental details in essential sections, and the study suffers from random study design decisions (e.g. N vs S titers). This impairs the usability of the data presented here, and makes it difficult put the presented results in context with other work. The overall writing is difficult to follow and much of the methodology is omitted, limiting a reader’s ability to follow any arguments or judge the strength of conclusions based on any given experiment.

[Response] We agree with the reviewer’s comments. We have provided more details for assays in the Methods section of the revised manuscript. We additionally conducted ELISA to quantify the anti-RBD and anti-S IgG antibody. The methods and results data have also been added in the Methods section (Line 447-460) and the Results section (Line 129-139) in the revised manuscript. The revised manuscript has been edited and refined throughout by a senior researcher with fluent English.

(5) As such, the primary value of this study lays in the observation that neutralizing antibody titers did not wash out as quickly as previously assumed. However, as with the their findings regarding T cell responses, the observed differences to other studies are not satisfactorily explained or discussed. The manuscript cannot be accepted in its current form and needs major revisions before it could be reconsidered.

[Response] Thanks! Please refer to our response to “Comment 3”. We have added more discussion of our T cell responses with other studies (Line 282-322).

(6) Manuscript: The manuscript would benefit from proof reading to smooth out the use of English, several sections are quite difficult to follow because of word usage. Also, it is written in a manner that assumes a high level of technical familiarity with various abbreviations, techniques, immune cell definitions (i.e. phenotypically effector memory (CD45RA⁻CCR7⁻) and effector (CD45RA⁺CCR7⁻) cells in response to both rAd5-S and rAd5-N rather than effector memory and central memory T cells” (no explanation of how the effector memory and central memory cells would differ from what they observe), not suitable for wide readership in Nat Comm.

[Response] The revised manuscript has been edited and refined by a senior researcher with fluent English throughout the manuscript. We have also made revisions for the abbreviations, techniques, and immune cell definitions throughout the revised manuscript. We have also discussed memory T cell response between we observed and others to clarify the differences in the revised manuscript (Line 327-340). This paragraph has been re-written as follow: ‘...Each subset of these T cells plays a role in the protective immunity to reinfection by rapidly migrating effector subsets into tissues to provide protection and proliferating central memory T cells in the draining lymph node to provide a pool of new effector cells(Glennie et al., 2015). Peng et al.(Peng et al., 2020) reported that SARS-CoV-2-specific CD8⁺ T cells among convalescent patients in the United Kingdom were mainly effector memory (CD45RA⁻CCR7⁻, 50.3±13.3%) and central memory (CD45RA⁺CCR7⁺, 20.7±8.4%) phenotypes. Zhou et al.

found similar trends for both CD4⁺ and CD8⁺ T cells responsive to SARS-CoV-2 N protein and RBD among COVID-19 convalescent patients in Hong Kong (Zhou et al., 2020). In our study, although varied frequencies of SARS-CoV-2-specific CD4⁺ and CD8⁺ T cells were detected in patients, the majority of S- and N-specific CD4⁺ and CD8⁺ T cells were phenotypically effector memory (CD45RA⁺CCR7⁺) and late effector (CD45RA⁺CCR7⁻) T cells. This phenome was also observed in COVID-19 patients about 1 month after infection(Weiskopf et al., 2020). It is possible that S- and N-specific T cells expressing central memory (CD45RA⁻CCR7⁺) phenotype fall off rapidly after the infection has resolved.'

(7) Methods: Do not clearly explain their methodologies in the results section, meaning one has to hope it is explained in the figure captions or methods and then piece together the information

[Response] Thanks! We have correspondingly made revisions in each section of results in the revised manuscript (Line 113-116, 129-131, 145-146, 172-177, 209-212).

(8) Figures: Figures are overloaded and text labels are often too small. The use of larger font-sizes would be advisable.

[Response] Thanks for your suggestions, we have made changes in the revised figures. In addition, because we included new data in the revised manuscript and to make the figure more logical, we split the original figure 1 into two figures, the new figure 1 focus on IgG and IgM antibodies response and new figure 2 focus on neutralizing antibodies response and correlation between studies antibodies.

(9) Sup. Data: Weak supporting data, one experimental setup for each claim, no considerations of alternate explanations or limitations of their experimental systems

[Response] Thanks, but the supplementary data for this manuscript just related the figures, and we have provided more details for the legend of supplementary figures. We have also considered different assays' limitations, so we used different assays to detect antibody response and used the full length of S and N for virus-specific T cell response. We have discussed this in the revised manuscript (Line 177-179, 267-272, and 315-317).

(10) Line 29: Clarify the wording of IgG and IgM, assume reference to SARS2-spike binding IgG & IgM? Or is it whole IgG & IgM in the serum?

[Response] Thanks! Corrected in the abstract of the revised manuscript.

(11) L55-58: referencing is inconsistent; refs 7,8,9 should also be referenced in the statement in line 55.

[Response] Thanks! Corrected (Line 64).

(12) Line 63: meaning not clear; perhaps 'or not present at all'?

[Response] Yes, we mean "not present at all patients", we have clarified in the revised manuscript (Line 75-76).

(13) Line 82: 'the same ... RBD antigen as the Ab-ELISA'? What is Ab-ELISA referring to?

[Response] Thanks for point this out. Assuming that the reviewer means Line 282, it is miswriting, and we have corrected this in the revised manuscript (Line 436).

(14) Line 93: how were IgG and IgM measured? Against what antigen? Not stated in the text.

[Response] Thanks! We used two commercial ELISA kits to detect IgG against nucleoprotein and IgM against RBD. We have included this information in the revised manuscript (Line 113-116).

(15) Line 95: It is not clear which antigen the titers were assessed. From the method section it could be both anti-nucleoprotein or anti-RBD antibodies. Please clearly indicate the parameters assessed here, including the serum dilutions used.

[Response] Thanks! Please refer to our response to 'Comment 14'. We have added detailed information for the IgG and IgM antibody detection in the methods section in the revised manuscript (Line 418-420). According to the manufacturer's instruction, the serum specimen dilution was 1:10. We have added this to the revised manuscript (Line 116, 425-426).

(16) Line 109: clarify that NAb was detected out of whole sera

[Response] Thanks! Yes, the NAb were detected in the serum, and we have clarified in the revised manuscript (Line 145).

(17) Line 110: define acronym GMT? What units does GMT have?

[Response] Thanks! GMT is the abbreviation of geometric mean titers and has been defined in the revised manuscript (Line 147-148). Because the geometric mean is calculated by averaging the \log_{10} transformed neutralizing antibody titer, such as 80, 160, and then converting the mean to a real number, so GMT has no units.

(18) Line 120: which Ig correlates better with Neut? IgG or IgM?

[Response] Because we added new results data of anti-S and RBD IgG, we re-analyzed the correlation between neutralizing antibody titers and IgG and IgM antibodies. We found all IgG and IgM antibodies correlated with neutralizing antibody titer. However, the anti-RBD IgG antibody had a better correlation with neutralizing antibodies. These results have been included in the revised manuscript (Line 159-165).

(19) Line 122: was IgG only measured against N (define acronym) and IgM against S (define acronym)? Can the authors provide supporting evidence that the IgG does not bind to S, or that IgM doesn't bind to N? It seems unlikely there would be an isotype specific antigen recognition related to variable domains of antibodies.

[Response] Yes, we only measured anti-nucleoprotein IgG and anti-RBD IgM in the initial manuscript. We have defined 'N' and 'S' throughout the revised manuscript. We agree with the reviewer that serum IgG and IgM antibodies can bind both spike and nucleoprotein. We apologize for not having made the correct understanding. We have

deleted the statement in the revised manuscript (Line 165-166). As addressed in our response to the second comment by the reviewer, we additionally conducted ELISA to quantify the anti-RBD and anti-S IgG antibody. The methods and results data have also been added in the Methods section (Line 447-460) and the Results section (Line 129-139) in the revised manuscript.

(20) Line 131: are control healthy donors also tested with the Ad5 recognition assay?

[Response] We did not test S and N specific T cell responses in healthy controls because we used Ad5-GFP as control for testing. We have clarified this in the Methods section of the revised manuscript (Line 479).

(21) Line 134: the authors measure T cell responses 3-4 months after recovery, yet claim based on the behaviour of these post-infection T cells that T cell function was impaired during infection. Can the authors support this statement with any further data? Can they, for instance, be certain that the S and N specific T cell response has not degraded over the 3-4 months after active infection?

[Response] We think the reviewer's comments very pertinent. We do not have any further data and, therefore, cannot just claim "T cell function was impaired during infection", and we have deleted this in the revised manuscript (Line 204-205). It is possible that S- and N- specific T cell response has degraded over the 3-4 months after infection. We have included such statement in the Discussion section (Line 294-295). Furthermore, we re-written these results in revised manuscript (Line 190-204) as below: *"We further measured the TNF- α co-expression in all the 14 patients with IFN- γ ⁺ CD4⁺ or CD8⁺ T cells and co-expression of GzmB in 8 of them (Fig. 3c). In response to rAd5-S, TNF- α co-expression with IFN- γ was detected in 7 of the 10 patients with S-specific CD4⁺ T cells and 2 of the 3 patients with S-specific CD8⁺ T cells, while GzmB co-expression with IFN- γ was found in 4 of the 6 patients examined with S-specific CD4⁺ T cells and one patient examined with S-specific CD8⁺ T cells (Fig. 3d, e). In response to rAd5-N, TNF- α co-expression with IFN- γ was detected in 11 of the 13 patients with N-specific CD4⁺ T cells and 6 of the 7 patients with N-specific CD8⁺ T cells, while GzmB co-expression with IFN- γ was found in 5 of the 7 patients examined with N-specific CD4⁺ T cells and 5 of the 6 patients examined with N-specific CD8⁺ T cells (Fig. 3d, e). Notably, the variation in the co-expression was dramatic. In most patients, less than 50% of IFN- γ ⁺ CD4⁺ and CD8⁺ T cells co-expressing TNF- α or GzmB were detected (Fig. 3d, e). A similar varied and overall low proportion of co-expression was observed in convalescent patients with COVID-19 in the UK by Peng et al. (Peng et al., 2020) and in Swedish by Sekine et al. (Sekine et al., 2020)."*

(22) Line 167: Why unexpectedly?

[Response] With "unexpectedly" we intended to express that because all studied patients remain positive for IgG and neutralizing antibody 3-4 months after infection, but not all patients had detectable T cell responses. Improper usage of the word "unexpectedly" caused a misunderstanding, and we have deleted it and made revision in the revised manuscript (Line 240).

(23) Line 181: 'Of interesting, in comparison with negative IgG detection in 12.9% of symptomatic patients 8 weeks after discharge, ...'. Meaning 'Interestingly, compared to 12.9% negative IgG levels published for symptomatic patients 8 weeks following discharge, ..'?

[Response] Thanks! Yes, we have revised this sentence in the revised manuscript (Line 259-264).

(24) Line 191: English incorrect

[Response] Thanks! More detailed discussion was made in the revised manuscript (Line 277-322).

(24) Line 194: Duplication of the word 'However'

[Response] Thanks! Corrected (Line 291).

(26) Line 205: Not clear what authors mean.

[Response] We have deleted this statement instead of "It is possible that S- and N-specific T cells expressing central memory (CD45RA⁻CCR7⁺) phenotype fall off rapidly after the infection has resolved" in the revised manuscript (Line 339-442).

(27) Line 277ff: It is not clear what was measured.

[Response] Thanks! We mean to measure the IgG and IgM antibodies in serum, and we have clarified in the revised manuscript (Line 418-421).

(28) Figure 1: It is not clear whether the IgM and IgG titers indicated are against the nucleoprotein or the RBD. Also, how are Nab titers presented in these graphs? What do the 1, 2, 3, 4 mean?

[Response] We have clarified in revised figure 1 and its legend, and throughout the revised manuscript. The Nab titers, such as 40, 80, and 160, were first log₁₀ transformed to a real number and presented in the figure. The numbers 1, 2, 3, and 4 are logarithmic geometric scale representing Nab titers 10, 100, 1000, and 10000, respectively.

(29) Figure 2: Why did the authors use SEM instead of SD? Is a change of IFN- γ ⁺ between 0.04% and 0.2% meaningful? How does the magnitude of the measures for these experiments compare with other work? How does it compare to healthy donors, do they react to any of the Ad5s?

[Response] Thanks! The reviewer is right. It should be SD, and we have revised this in the revised figure 3 and correspondingly figure legend. The IFN- γ ⁺ T cells are 0.04% in rAdv5-GRP response (controls), which is below the essentially at the positivity cutoff of 0.05% for a typical ICS assay, while 0.2% IFN- γ ⁺ T cells response with rAdv5-S or N indicates a virus-specific response. Please refer to our response to "Comment 3" by the reviewer, and we have discussed magnitude with other works in the revised manuscript (Line 282-322).

(30) Ext. Fig.1: The figure is full of white space, yet the information is printed so small that it barely can be read. Range of axes need to be adjusted to reflect the data represented and label and symbol sizes need to be increased.

[Response] Thanks! We have made changes to supplementary figures. Because we have added additional results of anti-S and -RBD IgG antibodies, the correlation of each isotype antibody with T cell response was present in one supplementary figure, and four supplementary figures (Supplementary Figs. 1-4) were produced following the reviewer's suggestion.

(31) Ext. Fig.2: Too low resolution bitmaps with percentages so small they cannot be read on a printout. It appears that the singlet gating on FSC-W/FSC-H and SSC-W/SSC-H graphs are not visible. Is the upper IFN- γ graph from CD8⁺ and the lower from CD4⁺? They are not labelled.

[Response] Thanks! We have re-produced the Ext. Fig. 2 (named Supplementary Fig. 5 in the revised manuscript). The invisible singlet gating on FSC-W/FSC-H and SSC-W/SSC-H in initial Extended. Fig. 2 may be caused by low resolution. It is clear that singlet gating on FSC-W/FSC-H and SSC-W/SSC-H in revised Supplementary Fig. 5. Yes, the upper is CD8⁺ T cells, and lower is CD4⁺ T cells, we have labeled as the reviewer suggested. In addition, we have included more details for the legend of the figure now.

Reviewer 2

(1) This study characterized serum antibody binding and neutralizing responses, and CD4 and CD8 T cell responses, to SARS-CoV-2 infection out to 3-4 months post-infection in a cohort of 25 patients. The median age in the cohort was 40 years, roughly half were male (13/25), and most (18/25) of the cohort had moderate infection. To this reviewer, the most important findings are that 100% of the cohort produced binding and neutralizing antibodies by 14 days post-infection (and >80% produced such responses by day 10); the neutralizing titers were high in a live-virus neut assay at 15-21 days post-infection (GMT = 1290; 95% CI 873-1875); and the neutralizing titers were relatively sustained out to three months post-infection, decaying by less than a factor of two (GMT = 697; 95% CI 401-1215). The manuscript also found that up to 65% of patients mounted detectable virus-specific CD4 or CD8 memory responses at the 3-4 month timepoint.

The relatively high durability of the neutralizing responses detected here are an important early indication that infection potentially might induce relatively well-sustained protective antibody responses, and these data also provide at least one important benchmark for ongoing vaccine studies. This reviewer believes that these are timely and important findings for the field to consider.

[Response] We thank the reviewer for this positive assessment.

(3) The section in lines 54-68 fails to cite or discuss many papers published characterizing neutralizing antibody responses to SARS-CoV-2. For example: Brouwer, van Gils et al Science 2020; Cao, Xie et al Cell 2020; Hansen, Kyratsous et al Science 2020; Ju, Zhang et al Nature 2020; Kreer, Klein et al Cell 2020; Robbiani, Nussenzweig et al Nature 2020;

Rogers, Burton et al Science 2020; Seydoux et al Immunity 2020; Wec, Walker et al Science 2020; Wu, Liu et al Science 2020.

[Response] Thanks for your comments. We cited these references in the revised manuscript (Line 66-68, 71).

(4) lines 135-137 has a typo and conflicts with lines 190-192. lines 135-137 say “approximately 65% of recovered patients could not produce SARS-CoV-2 specific CD4 or CD8 T cell response 3-4 months after infection”, while lines 190-192 say “We found that only approximately 65% of patients had detected SARS-CoV-2-specific CD4⁺ or CD8⁺ T cells 3-4 months after infection.”

[Response] Thanks for pointing this inconsistency. It should be ~65% of recovered patients had detectable virus-specific T cell response 3-4 months after infection. We have included more detailed discussion in the revised manuscript (Line 277-322).

(5) The CD4 and CD8 response rate (% responders) is not clearly indicated in Fig. 2, although the authors give such response rates in the text. It would be helpful to add information or a panel to this figure giving the response rates, similar to the “positive rate” shown in Fig 1A,B for antibody responses.

[Response] Thanks for this useful suggestion, and we have added the response rate in the revised figure 3.

(6) It is unclear to this reviewer if the detection of CD4 or CD8 responses responses in 65% or less of patients might or might not be due to technical limitations of the assay used. Were prior analyses of cellular responses to SARS-CoV-2 that showed higher response rates at 1 month post-infection carried out using a similar or different assay?

[Response] We think the reviewers' comment very pertinent. We did not analyze cellular responses to SARS-CoV-2 at one month post-infection. Therefore, we cannot rule out the possibility that the detection of CD4 or CD8 responses in 65% of patients might be technical limitations of the assay. We have included such statements in the Discussion section (Line 291-295) as following: " *However, Howeverthe potential factors for such differences cannot be ruled out the technical limitation of the assays used because we do not have enough controls. Moreover, it is possible that HLA genotypes playing an important role. Another potential possibility is that S-specific T cells, especially for CD8⁺ T cell responses, have degraded over the 3-4 months period.*"

REVIEWERS' COMMENTS

Reviewer #1 (Remarks to the Author):

The authors have addressed most individual points of criticism, and the quality of the manuscript has substantially improved. However, the main concerns regarding study design, in particular the use of different antigens to assess the different isotypes, remain unaddressed. Also, the longevity of the antibody response constitutes a highly controversial subject; the authors offer the use of different assays as possible explanation for contradictory findings, which I do not believe (and which frankly would be highly concerning if it was actually the case). My enthusiasm for the study therefore remains somewhat limited, even after the substantial improvements.

Reviewer #2 (Remarks to the Author):

The authors have done a good job at addressing the questions of both reviewers. The manuscript has been substantially improved in many dimensions, and it remains interesting and timely.

There are still a few English grammar/usage problems (e.g. two errors in lines 239-240) but I assume these will be cleaned up with the help of NatComms editorial staff in the proofs.

Response to reviewers' comments

Reviewer 1 (Remarks to the Author)

(1) The authors have addressed most individual points of criticism, and the quality of the manuscript has substantially improved. However, the main concerns regarding study design, in particular the use of different antigens to assess the different isotypes, remain unaddressed. Also, the longevity of the antibody response constitutes a highly controversial subject; the authors offer the use of different assays as possible explanation for contradictory findings, which I do not believe (and which frankly would be highly concerning if it was actually the case). My enthusiasm for the study therefore remains somewhat limited, even after the substantial improvements.

[Response] We appreciate the reviewer comments. The two structural proteins of SARS-CoV-2, nucleocapsid (N) and spike (S) protein, have been used as target antigens for serological assays. Although it is unlikely that antibody responses to N protein can directly neutralize SARS-CoV-2, this is the antigen targeted by multiple commercial assays. Therefore, to study antibody responses to SARS-CoV-2, we first measured IgG against N protein and IgM against the receptor-binding domain (RBD) of the SARS-CoV-2 S protein in serum at a 1:10 dilution from the patients by two well-validated commercial diagnostic ELISA kits. We then focused on IgG antibodies to the spike trimer and the RBD monomer since the neutralizing antibody (NAb) response for SARS-CoV-2 primarily targets the S protein. We have included such statement in the revised manuscript (Line 107-110).

Although few studies of antibody response to SARS-CoV-2 in recovered patients approximately 3-4 months were controversial with our and other studies we mentioned in the manuscript, more and more evidence showed that the durability of SARS-CoV-2 antibody responses is relatively stable 3-5 months after infection (Crawford et al., 2020; Grandjean et al., 2020; Gudbjartsson et al., 2020; Isho et al., 2020; Iyer et al., 2020; Seow et al., 2020; Wajnberg et al., 2020a). As the reviewer concerning, we have revised the sentence involving different assays for explanation for contradictory findings in the revised manuscript (Line 251-254). We believe our findings are important to understand the persistence of immune response induce by SARS-CoV-2 infection and the lasting immune response is important for controlling the COVID-19 pandemic and vaccine development.

Reviewer 2 (Remarks to the Author)

(1) The authors have done a good job at addressing the questions of both reviewers. The manuscript has been substantially improved in many dimensions, and it remains interesting and timely.

[Response] We appreciate the reviewer's positive assessment of our study.

(2) There are still a few English grammar/usage problems (e.g. two errors in lines 239-240) but I assume these will be cleaned up with the help of NatComms editorial staff in the proofs.

[Response] Thanks for point these two typos out. We have carefully edited the language

throughout of the revised manuscript in addition to correct these two typos (Line 254-256).